# Antibacterial Property of Composites of Reduced Graphene Oxide with Nano-Silver and Zinc Oxide Nanoparticles Synthesized Using a Microwave-Assisted Approach

**DOI:** 10.3390/ijms20215394

**Published:** 2019-10-29

**Authors:** Yi-Huang Hsueh, Chien-Te Hsieh, Shu-Ting Chiu, Ping-Han Tsai, Chia-Ying Liu, Wan-Ju Ke

**Affiliations:** 1Department of Seafood Science, National Kaohsiung University of Science and Technology, Kaohsiung 81157, Taiwan; 2Department of Chemical Engineering and Materials Science, Yuan Ze University, Taoyuan 32003, Taiwan; qcricnt@hotmail.com; 3Graduate School of Biotechnology and Bioengineering, Yuan Ze University, Taoyuan 32003, Taiwan; n8808701@gmail.com; 4Department of Internal Medicine, Far Eastern Memorial Hospital, New Taipei City 220, Taiwan; chiaying.liu@gmail.com; 5Graduate Institute of Biomedical Sciences, and Research Center for Bacterial Pathogenesis, Chang Gung University, Taoyuan 33302, Taiwan; d8901005@stmail.cgu.edu.tw

**Keywords:** antibacterial agents, *Escherichia coli*, graphene, minimum inhibitory concentration, nanocomposite, nano-silver, *Staphylococcus aureus*, sterilization

## Abstract

Graphene oxide (GO) composites with various metal nanoparticles (NPs) are attracting increasing interest owing to their broad scope in biomedical applications. Here, microwave-assisted chemical reduction was used to deposit nano-silver and zinc oxide NPs (Ag and ZnO NPs) on the surface of reduced GO (rGO) at the following weight percentages: 5.34% Ag/rGO, 7.49% Ag/rGO, 6.85% ZnO/rGO, 16.45% ZnO/rGO, 3.47/34.91% Ag/ZnO/rGO, and 7.08/15.28% Ag/ZnO/rGO. These materials were tested for antibacterial activity, and 3.47/34.91% Ag/ZnO/rGO and 7.08/15.28% Ag/ZnO/rGO exhibited better antibacterial activity than the other tested materials against the gram-negative bacterium *Escherichia coli* K12. At 1000 ppm, both these Ag/ZnO/rGO composites had better killing properties against both *E. coli* K12 and the gram-positive bacterium *Staphylococcus aureus* SA113 than Ag/rGO and ZnO/rGO did. RedoxSensor flow cytometry showed that 3.47/34.91% Ag/ZnO/rGO and 7.08/15.28% Ag/ZnO/rGO decreased reductase activity and affected membrane integrity in the bacteria. At 100 ppm, these two composites affected membrane integrity more in *E. coli*, while 7.08/15.28% Ag/ZnO/rGO considerably decreased reductase activity in *S. aureus*. Thus, the 3.47/34.91% and 7.08%/15.28% Ag/ZnO/rGO nanocomposites can be applied not only as antibacterial agents but also in a variety of biomedical materials such as sensors, photothermal therapy, drug delivery, and catalysis, in the future.

## 1. Introduction

Graphene oxide (GO) has six carbon atoms of graphene with several oxygen functional groups such as the hydroxyl (-OH), epoxy (-CH-O-CH), and carboxyl (-C-OH) groups. It is a two-dimensional planar structure with a thickness of a single layer to a few layers [1,2,3]. GO and its derivatives have been applied in soft electronics, functional composites, energy storage materials, medical equipment, gas sensor, and drug transportation [4,5,6,7,8,9,10].

Controlling the degree of reduction of GO can indirectly control the energy gap, thereby regulating its optical properties and causing different optical absorption and optical response characteristics in different spectral regions [11,12]. Because of its optical confinement properties, GO can protect sensitive instruments from laser-induced damage, and it is, therefore, used as an optical nonlinear material [13]. In addition, GO is easily dispersed in water and decomposed into macroscopic sheets. These sheets undergo chemical reduction to produce graphene. Therefore, GO is a very good material for the large-scale production of graphene. The recent synthesis of GO composites with various metal nanoparticles (NPs) has attracted wide interest owing to their potential applications in a range of biomedical materials [14,15,16,17,18]. Due to their high surface area, GO composites support the growth and stability of metal NPs and prevent them from agglomerating. Some recent studies have focused on the antibacterial activity of GO. However, research on the antibacterial activity of graphene and metal nanocomposites has been rather limited.

Many metal oxide complex, metallic particles were used as antibacterial agents [19,20,21] and nano-silver and zinc oxide are most common used [21]. Recently, several studies have focused on the nano-silver oxide–graphene and nano- ZnO–graphene complexes because they have better antimicrobial and antifungal activities than those of GO alone [22,23,24,25,26,27,28]. The application of silver/zinc oxide/reduced graphene oxide (Ag/ZnO/rGO) nanocomposites in bacterial detection and sterilization has also been studied [29,30]. The Ag/ZnO/rGO nanocomposite has been found to function in bacterial detection and sterilization, opening up a new angle to study the antibacterial efficacy of different composites of nanomaterials and GO. However, thus far, minimum inhibitory concentration (MIC) experiments for such nanomaterial composites with GO against bacteria have been limited, and the exact mechanisms underlying the antibacterial effects remain to be deciphered. In this study, microwave-assisted chemical reduction was used to deposit Ag and ZnO on the surface of graphene. We used different ratios of Ag/ZnO/rGO nanocomposites to test their effects on the gram-negative bacterium *Escherichia coli* and the gram-positive bacterium *Staphylococcus aureus* with the aim to obtain a clearer picture of the mechanisms involved.

## 2. Results

### 2.1. Synthesis and Morphological Analysis of Ag/rGO, ZnO/rGO, and Ag/ZnO/rGO Composites

Ag/rGO, ZnO/rGO, and Ag/ZnO/rGO were synthesized as described in the methods section. The synthesized composites, along with their weight% values, are presented in Table 1. The X-ray photoelectron spectroscopy (XPS) analysis data in Table 1 show the integrated area percentages of Ag/rGO, ZnO/rGO, and Ag/ZnO/rGO to be 5.34%, 7.49%, 6.85%,16.45%, 3.47/34.91%, and 7.08/15.28%, respectively (the synthesis weight% did not present the final percentage after coating onto rGO surfaces). The XPS analysis revealed a slight difference in the chemical composition of as-prepared samples, compared to the recipe of the samples prepared (see Table 1). This difference can be attributed to two possible reasons. First, Ag^2+^ and Zn^2+^ ions display different equilibrium adsorption capacities onto rGO sheets before chemical reduction under microwave irradiation. The consumption of ionic adsorption should reach ~100%, approaching the chemical ratio of sample preparation. However, it is difficult to attain the situation (i.e., 100% ionic adsorption) due to porous structure and surface heterogeneity (e.g., oxygen functional groups) in rGO sheets.

We use these designated percentages hereafter. The representative scanning electron microscopy (SEM) images of the Ag/rGO, ZnO/rGO, and Ag/ZnO/rGO composites are shown in Figure 1a–f. A large GO sheet of nanometer dimensions was found to be situated on top of the grid (Figure 1a–f), with a transparent and rippled silk wave appearance. The wrinkles and folded parts are attributed to GO. In the SEM image of Ag/rGO and ZnO/rGO (Figure 1a–d), the small spots dispersed on the GO flake are attributed to the Ag or ZnO NP composites on the nanosheet of rGO. X-ray diffraction (XRD) verified rGO by the reappearance of the (002) diffraction peak at 25.5 [31]. The main peaks of the XRD pattern of the Ag/ZnO nanocomposite are shown in Figure 2a. The diffraction peaks of the Ag/ZnO nanocomposite are similar to those of Ag NPs. The four obvious diffraction peaks located at 38.1, 44.3, 64.4, and 74.4 in the XRD pattern of the 5.34% and 7.49% Ag/rGO composites are assigned to (111), (200), (220), and (311) of the crystalline planes of Ag, suggesting that Ag NPs exist on the rGO sheet (as refer in JCPDS Card No. 04-0783 [32]). Ag/rGO and ZnO/rGO composites had this peak, but Ag/ZnO/rGO did not (Figure 2a,b). Comparisons with the typical ZnO NPs in the XRD pattern are shown in Figure 2b; 16.45% ZnO/rGO had the peaks at 32, 34.6, 36.4, 47.8, 56.8, 63, and 68.3, corresponding to (110), (002), (101), (102), (110), (103), and (112) crystalline planes of ZnO (as refer in JCPDS Card No. 36-1451 [33]), respectively, and 6.85% ZnO/rGO had no sharp peaks and fewer peaks at 47.8 (102), 63 (103), and 68.3 (112) [34]. This might be due to a lower amount of ZnO NPs. Moreover, 3.47/34.91% Ag/ZnO/rGO and 7.08/15.28% Ag/ZnO/rGO did not have clear peaks for Ag or ZnO, which might be ascribed to interference. The size distribution of Ag and ZnO NPs are shown in Figure 3a–d. The particle size distribution of each sample was obtained by counting more than 100 nanoparticles in the field-emission scanning electron microscopy micrograph. The raw data were then normalized for the particle size distribution analysis that determines the average size and range of particles representative of the given material. For the 5.34% and 7.49% Ag/rGO composites, Ag NPs dispersed differently from each other, and over 50% Ag NPs were <10 nm in size (Figure 3a,b). ZnO NPs dispersed differently, and over 50% ZnO NPs were <10 nm in size (Figure 3c,d).

### 2.2. XPS Analysis

Figure 4a,c,k,o shows the XPS signature of the Ag 3d doublet (3d5/2 and 3d3/2) for the Ag NPs deposited on rGO. The Ag 3d5/2 and 3d3/2 peaks of Ag/rGO nanocomposites appeared at around 367.93 and 373.8 eV, respectively, which are characteristic peaks for silver metal [25]. As illustrated in the upper panels of Figure 4b,g,j,n,r the XPS spectrum of rGO at 284.2 eV showed the characteristic peaks of C–C and C=C [35]. In addition, two diffraction peaks appeared at ca.1021.5 eV and ca. 1045.0 eV, corresponding to Zn 2p1/2 and Zn 2p2/3, respectively, for ZnO/rGO (Figure 4e,h,l,p) [36]. The O 1s spectrum showed a highly intense peak at approximately 531.7 eV, which can be ascribed to lattice oxygen with an oxidation state of O^−2^ in the ZnO (Figure 4f,i,m,q) [37].

### 2.3. Fourier-Transform Infrared Analysis

GO and the Ag/rGO, ZnO/rGO, and Ag/ZnO/rGO nanocomposites had the typical D and G peaks (data not shown) in Fourier-transform infrared (FTIR) analysis. Figure 5 shows the FTIR spectra of GO and the Ag/rGO, ZnO/rGO, and Ag/ZnO/rGO nanocomposites. All compounds had the C=O carbonyl stretching peak at 1723 cm^−1^, the C–O epoxide group stretching peak at 1178 cm^−1^, and the peak at 1620 cm^−1^ related to the skeletal stretching of the C=C alkene group, similar to previous observations [38]. The rGO layer contained some oxygen functional moieties, and the peak located around at 2920 cm^−1^ was assigned to the stretching of C–H in the methyl group [39]. A small peak observed at 890 cm^−1^ could be attributed to the =C–H of plane bending from the alkene group [26]. The peak observed at around 3620 cm^−1^ could be attributed to the -OH group. Neither Ag/rGO nor GO had C–H or -OH bonds, but ZnO/rGO and Ag/ZnO/rGO did. Thus, the binding of ZnO NPs, but not Ag NPs, onto the rGO surface may produce -OH or C–H bonds.

### 2.4. The Ag/ZnO/rGO Nanocomposite Inhibited Bacterial Growth

We sought to assess the effects of different ratios of Ag/rGO, ZnO/rGO, and Ag/ZnO/rGO against gram-negative *E. coli* K12 and gram-positive *S. aureus* SA113 to elucidate the toxicity mechanisms involved. We treated *E. coli* K12 and *S. aureus* SA113 cultures with 0–1000 ppm of the six nanocomposites and evaluated bacterial growth over 24 h. In the MIC test for *E. coli*, at 100 ppm, 5.34% Ag/rGO, 7.49% Ag/rGO, 3.47/34.91% Ag/ZnO/rGO, and 7.08/15.28% Ag/ZnO/rGO had better inhibition of around 4-log CFU reduction compared to the ZnO/rGO composites (Figure 6a). At 1000 ppm, 5.34% Ag/rGO caused a 5-log reduction in CFU, while 7.49% Ag/rGO, 3.47/34.91% Ag/ZnO/rGO, and 7.08/15.28% Ag/ZnO/rGO killed around 9-log CFU, but ZnO/rGO showed a much lower killing effect (Figure 6a). This suggests that ZnO embedding onto rGO did not enhance the killing effects. For *S. aureus* SA113, at 100 ppm, 5.34% Ag/rGO, 7.49% Ag/rGO, 3.47/34.91% Ag/ZnO/rGO, and 7.08/15.28% Ag/ZnO/rGO had better inhibition of around 1-log CFU reduction compared to the ZnO/rGO composites. At 1000 ppm, 5.34% Ag/rGO, 7.49% Ag/rGO, 7.08/15.28% Ag/ZnO/rGO, and 3.47/34.91% Ag/ZnO/rGO showed reductions of 2-, 3-, 5-, and 9-log CFU, respectively, but ZnO/rGO composites did not have sufficient killing ability (Figure 6b). Again, ZnO embedded onto rGO did not increase the bactericidal effect against *S. aureus*. However, at the high concentration of 1000 ppm, both 3.47/34.91% Ag/ZnO/rGO and 7.08/15.28% Ag/ZnO/rGO had an enhanced killing effect against both bacteria compared to Ag/rGO only.

### 2.5. The Ag/ZnO/rGO Nanocomposite Altered the Redox Status and Membrane Integrity in S. aureus and E. coli

We examined whether 3.47/34.91% and 7.08/15.28% Ag/ZnO/rGO affected reductase activity in *S. aureus* and *E. coli* by RedoxSensor staining. Ag/ZnO/rGO composites were found to reduce the geometric mean of reductase activity. The geometric mean values of reductase activity in *S. aureus* and *E. coli* were 18,104 and 12,390 arbitrary units (a.u.), respectively, for no treatment (0 ppm). The percentages of gated cells in the total cell population were 99.7% for *S. aureus* and 98.2% for *E. coli* (Figure 7). With 100 ppm rGO, the geometric mean values of reductase activity in *S. aureus* and *E. coli* were 30,314, and 10,336 a.u., respectively, and the respective percentages of gated cells in the total cell population were 98.7% and 97.0%. With 3.47/34.91% and 7.08/15.28% Ag/ZnO/rGO at 100 ppm, the geometric mean values of reductase activity in *S. aureus* and *E. coli* were 7187 and 3094, and 4084 and 5225 a.u., respectively. The percentages of gated cells in the total cell population were 25.8% and 4.6%, and 17.2% and 24.0% for *S. aureus* and *E. coli*, respectively (Figure 7). The decrease in the percentage of gated *S. aureus and E. coli* cells positive for RedoxSensor staining under 100 ppm treatment with both 3.47/34.91% and 7.08/15.28% Ag/ZnO/rGO might be indicative of a decrease in cell reductase.

Furthermore, both 3.47/34.91% and 7.08/15.28% Ag/ZnO/rGO severely compromised the integrity of cell membranes after just 3 h of treatment and dramatically increased the percentage of cells stained with propidium iodide (PI). Without treatment (0 ppm), the geometric mean values of PI staining in *S. aureus* and *E. coli* were 16,387 and 18,581 a.u., respectively, and the respective percentages of gated cells in the total cell population were 3.8% and 3.9% (Figure 8). In addition, the geometric mean values of the PI staining in *S. aureus* and *E. coli* treated with 100 ppm rGO were 17,497 and 18,357, and the percentages of gated cells in the total cell population were 3.7% and 4.0%, respectively. With 3.47/34.91% and 7.08/15.28% Ag/ZnO/rGO at 100 ppm, the geometric mean values of the PI staining in *S. aureus and E. coli* were 13,462 and 14,365, and 9318 and 10,155, respectively. The percentages of gated cells in the total cell population were 14.2% and 7.7%, and 17.8% and 21.2%, respectively (Figure 8).

The high levels of PI in *E. coli* were consistent with the MIC data in Figure 6. This suggested that 3.47/34.91% and 7.08/15.28% Ag/ZnO/rGO more effectively attached to the cell membranes of *E. coli*. Moreover, 7.08/15.28% Ag/ZnO/rGO had a better negative effect on reductase activity for *S. aureus* but a lower effect on membrane integrity compared to 3.47/34.91% Ag/ZnO/rGO. Therefore, there is no direct relationship between reductase and membrane integrity. However, 3.47/34.91% Ag/ZnO/rGO did kill more in *S. aureus* at 1000 ppm, as demonstrated by a larger percentage of PI-stained cells. This suggested that 3.47/34.91% Ag/ZnO/rGO had a better bactericidal effect against *S. aureus* by affecting membrane integrity.

## 3. Discussion

Perreault et al. used chemical methods to produce a film composite of GO imparting antibacterial properties to the surface of the film without altering the inherent water and salt permeability [16]. The GO average sheet area, ranging from 0.01 to 0.65 μm^2^, was examined in terms of antimicrobial activity. The antimicrobial activity of GO surface coatings increased fourfold when the GO sheet area decreased from 0.65 to 0.01 μm^2^, indicating better antimicrobial effect by smaller GO sheets. They pointed out that GO deactivates bacteria via direct contact, inducing cell membrane damage due to reactive oxygen species arising due to charge transfer.

Nanda et al. used Raman spectroscopy to study the antibacterial activity of GO against the gram-negative bacterium *E. coli* and gram-positive *Enterococcus faecalis* [40]. They found that for *E. coli*, the MIC was 1 μg/mL, and that for *E. faecalis* was 4 μg/mL. They also observed that Raman peaks had higher concentrations of proteins and adenine in *E. coli* and *E. faecalis* after treatment with GO. SEM showed structural and morphological changes in *E. faecalis* and *E. coli* as the concentration of GO increased. The cell membrane of *E. coli* clearly exhibited damage, and SEM images showed that GO penetrated *E. coli* cells. The results suggested that GO penetrated cells through its sharp edges. When the concentration of GO increased, the cell membranes of both gram-positive and gram-negative bacteria were ruptured. Thus, their findings showed that the antibacterial mechanism against gram-positive and gram-negative bacteria was the same.

The nano-silver oxide–graphene (Ag/GO) complex has better antimicrobial and antifungal activities than those of GO itself. GO nanocomposites and carbon nanoscrolls (CNSs) filled with Ag NPs have more marked inhibition zones than only GO in studies on fungi. The MIC of CNSs–Ag NPs against *Candida albicans* was found to be 0.25 mg/mL, and the MIC of Ag/GO was 0.5 mg/mL. SEM images showed that the morphology of *C. albicans* and *Candida tropicalis* changed significantly after contact with CNSs–Ag NPs, and that major damage occurred in the cytoplasmic membrane of *C. albicans* [41].

Tang et al. [42] examined Ag/GO NPs with different ratios of Ag NPs and GO to study the antimicrobial activity against *E. coli* and *S. aureus*. They found that all nanocomposites except Ag/GO at 0.65:1 significantly reduced the viability of *E. coli* and *S. aureus* cells at concentrations less than 10 μg/mL; as the dose increased, the viability of the cells decreased. Among three Ag/GO ratios (1:1, 0.65:1, and 2:1), 1:1 showed the strongest antibacterial effect at a concentration of 2.5 μg/mL, reducing 40% of *E. coli* and *S. aureus* cells. They showed that the ratio of nano-silver and GO is important for the antibacterial activity of Ag/GO. They also showed that the bactericidal effect of nano-silver coated onto GO is due to the destruction of the cell membrane of *E. coli* and the inhibition of *S. aureus* cell division.

Ko et al. [29] studied the application of Ag/ZnO/rGO nanocomposites in bacterial detection and sterilization. To check the bacterial killing performance of Ag/ZnO/rGO nanocomposites, *E. coli* ATCC-25922 suspensions (10^7^ CFU/mL) in the dark, at near-infrared (NIR) illumination (0.69 W/cm^2^), or with a CW 808 nm diode laser, in the presence or absence of Ag/ZnO/rGO nanocomposites (0.2 mg/mL), were analyzed. After X-ray irradiation but without the nanocomposites, *E. coli* activity remained at 100% in the dark but decreased with NIR and full Xe lamp exposure. With Ag/ZnO/rGO, the percentage of survival decreased significantly, to 4.3%, 0.1%, and 0%, respectively, in the dark, under NIR exposure, and Xe lamp illumination. The presence of Ag/ZnO/rGO nanocomposites resulted in more effective killing of *E. coli* in all three conditions. In the dark, the killing of *E. coli* was mainly due to Ag NPs. The killing of *E. coli* by NIR or Xe illumination was affected not only by Ag NPs but also by photo-thermal ablation induced by rGO or ZnO. It is clear that the combination of the two mechanisms led to more effective bacterial killing. We found that 7.49% Ag/rGO, 3.47/34.91% Ag/ZnO/rGO, and 7.08/15.28% Ag/ZnO/rGO composites had at least 10-fold higher killing effect than *E. coli*-killing effect reported by Ko et al. [29].

## 4. Materials and Methods

### 4.1. Cell Growth Conditions

*E. coli* K12 *and S. aureus* SA113 [38] were maintained in Luria–Bertani (LB) medium or plated on 1.5% Bacto agar plates supplemented with LB medium at 37 °C. For the antimicrobial activity assay, overnight bacterial cultures of approximately 1 × 10^7^ CFU/mL were added in 25 mL Mueller Hinton broth (MHB) in 250 mL Pyrex flasks [27]. Ag/rGO, ZnO/rGO, and Ag/ZnO/rGO nanocomposites were added to achieve final concentrations of 0, 10, 100, or 1000 ppm. Bacterial cells were grown for 24 h at 37 °C, with agitation at 175 rpm. After 24 h inoculation, cultures were serially diluted, plated on LB agar plates, and incubated overnight at 37 °C. Cell numbers were counted directly after 18 h incubation. All experiments were performed twice separately, and each value represents the mean of three technical replicates. Statistical significance was analyzed by one-way analysis of variance (ANOVA).

### 4.2. GO Preparation

Natural graphite (25 g) and sodium nitrate (12.5 g) were slowly added into 575 mL sulfuric acid and stirred with a magnetic stirrer in an ice bath for 3 h until the temperature reached 0 °C. Then, 75 g potassium permanganate was slowly added, and stirring was continued at 0 °C for 3 h. With a reflux device and a thermostat, the temperature went from 0 °C to 35 °C. The overall reaction is exothermic. When the temperature reached 65–85 °C, it was held for about 30 min. After the temperature dropped to 35–45 °C, deionized water was slowly added. The reaction is still exothermic at this stage, so the temperature continues to rise. When the reaction temperature no longer rose (about 70–80 °C), the thermostat was turned on for heating up to 95 °C and maintained at this temperature for 3 h. Finally, hydrogen peroxide (30 wt%) was added, and the temperature was kept constant at 95 °C for 1 h. This was followed by replacing the ions, washing with deionized water, and centrifugation. Finally, the solution was put in a vacuum oven at 80 °C to complete the formation of GO powder.

### 4.3. Preparation of GO Nanocomposites with Ag and ZnO

The composites of nanometals and rGO were prepared by a chemical reduction method. We synthesized 7.5% Ag/rGO, 12% Ag/rGO, 7.5% ZnO/rGO, 12% ZnO/rGO, 7.5%/7.5% Ag/ZnO/rGO, and 3%/12% Ag/ZnO/rGO composites as follows. GO was slowly added to a container with ethylene glycol, which was placed on a magnetic stirrer and stirred for 1 h. Then, silver nitrate mixed with ethylene glycol was slowly added, followed by the addition of zinc acetate mixed with ethylene glycol. Stirring and mixing was continued for 1 h. Finally, sodium hydroxide mixed with ethylene glycol solution was added, and stirring and mixing were continued for 1 h. The uniformly mixed solution was placed in a microwave oven, subjected to a chemical reduction reaction, heated for 1 min, stirred for 3 min, and subjected to four cycles of this procedure. This was followed by placing on a magnetic stirrer and stirring to room temperature. The solution was subjected to suction filtration and rinsed 10 times with deionized water to neutrality. The upper layer of nanometal/rGO powder was collected and placed in an oven at 80 °C until complete drying. The nanometal/rGO composite material was placed at room temperature and ground to a powder using an agate mortar.

### 4.4. XPS Analyses

The chemical composition on the surface of the as-synthesized GO and Ag NP- and ZnO NP-doped GO composites were determined by XPS (Physical Electronic ESCA PHI 1600; Chanhassen, MN, USA) at an excitation energy of Al K_α_(1486.6eV). The C 1s (284.5 eV) peak was used as the calibration standard for the wide-region spectra of these samples with different C–C valences. The C 1s spectra can be convoluted into five peaks centered at 284.5 eV (C=C or C‒C), 285.2 eV (C‒N), 286.5 eV (C‒OH), 288.5 eV (C=O), and 289.5 eV (O‒C=O), respectively, indicating various types of C-bonding. XPS signals of the above species were recorded with a cylindrical mirror analyzer. The fractions of these samples with different C–C valences were calculated by their integrated peak areas. All measurements were performed in two separate experiments.

### 4.5. XRD Characterization

A X-ray diffraction (XRD, Shimadzu labx XRD-6000) spectroscope was used to characterize crystalline structures of carbon composites. The XRD patterns of the synthesized GO and Ag NP- and ZnO NP-doped GO composites were recorded at a scan rate of 4 degree/min using monochromatic Cu-Kα radiation (MXP18; MAC Science Co., Tokyo, Japan) at 30 kV and 20 mA. The recorded specific peak intensity and 2*θ* values were further identified using a database system (JCPDS).

### 4.6. Field Emission Scanning Electron Microscopy

The morphology, dispersion, and particle size of the Ag/ZnO NPs on rGO support were investigated by a field emission scanning electron microscope (FE-SEM, Rigaku, Tokyo, Japan). We also performed electron microscopy using a RU-H3R FE-SEM to characterize the prepared samples in terms of their structure and elemental composition.

### 4.7. FTIR

FT-IR spectra of GO and Ag NP- and ZnO NP-doped GO composites were obtained using a Bruker Tensor 27 FT-IR spectrometer with 32 scans in a frequency range of 4000 to 500 cm^−1^.

### 4.8. Reductase Activities

The reductase activities of *S. aureus* SA113 and *E. coli* K12 were determined using a BacLight™ RedoxSensor™ Green Vitality Kit (Thermo Fisher, Waltham, MA, USA) [43,44]. Overnight cultures of approximately 1 × 10^7^ CFU/mL in 25 mL of MHB were supplemented with the indicated concentrations of rGO for 3 h at 175 rpm and 37 °C. Cells were washed with 1X PBS buffer twice and diluted 10-fold with the same buffer. For *E. coli* K12, 1 µL of RedoxSensor™ Green reagent and 1 µL PI were added to the mixture. For *S. aureus* SA113, 1 µL of 10-fold diluted RedoxSensor™ Green reagent and PI dilution were added to the mixture. This was followed by incubation in the dark at room temperature for 5 min for the assessment of cell membrane integrity. Stained cells (1 mL) in PBS were assayed by flow cytometry using an Attune™ NxT Flow Cytometer (Thermo Fisher). Fluorescence filters and detectors were all standardized with green fluorescence collected in the FL1 channel (530 ± 15 nm) and red fluorescence collected in the FL3 channel (>650 nm). Data were analyzed using Attune™ NxT Flow Cytometer software.

## 5. Conclusions

Ag/rGO, ZnO/rGO, and Ag/ZnO/rGO nanocomposites were synthesized via rapid microwave irradiation, whereby Ag, ZnO, Ag/ZnO NPs were deposited uniformly on the surface of rGO. We found that two Ag/ZnO/rGO composites had a better bactericidal effect compared to Ag/rGO or ZnO/rGO only. These two Ag/ZnO/rGO composites affected membrane integrity more in the gram-negative bacterium *E. coli* than in the gram-positive bacterium *S. aureus*. Meanwhile, the 7.08/15.28% Ag/ZnO/rGO composite remarkably decreased the reductase activity in *S. aureus*. These two Ag/ZnO/rGO composites open up avenues for use as effective antimicrobials and further applications in various medical materials.

## Figures and Tables

**Figure 1 ijms-20-05394-f001:**
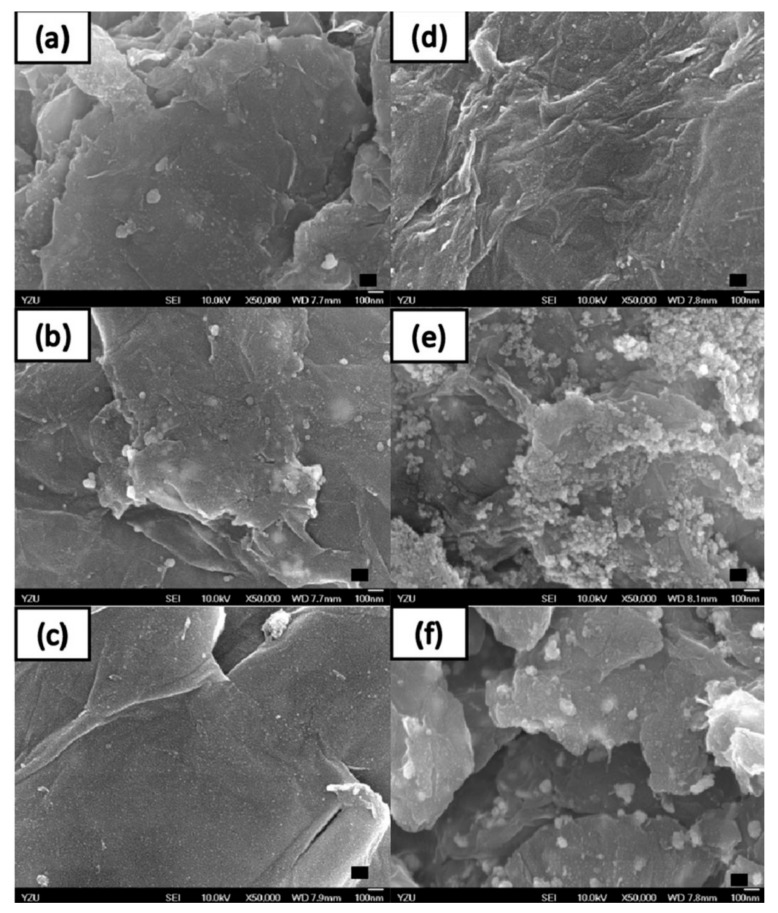
High-resolution scanning electron microscopy. Scanning electron microscopy images of (**a**) 5.34% silver nanoparticle-embedded graphene oxide (Ag/rGO), (**b**) 7.49% Ag/rGO, (**c**) 6.85% zinc oxide nanoparticle-embedded graphene oxide (ZnO/rGO), (**d**) 16.45% ZnO/rGO, (**e**) 3.47% silver nanoparticle, and 34.91% zinc oxide nanoparticle-embedded graphene oxide (Ag/ZnO/rGO), and (**f**) 7.08% silver nanoparticle and 15.28% zinc oxide nanoparticle-embedded graphene oxide (Ag/ZnO/rGO). Black bar is 100 nm. Data are representative of two separate experiments.

**Figure 2 ijms-20-05394-f002:**
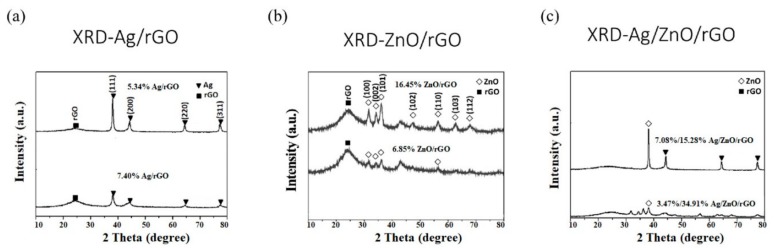
X-Ray diffraction analysis. (**a**) Silver nanoparticle-embedded graphene oxide (Ag/rGO), (**b**) zinc oxide nanoparticle-embedded graphene oxide (ZnO/rGO), and (**c**) silver nanoparticle and zinc oxide nanoparticle-embedded graphene oxide (Ag/ZnO/rGO). Black square (■), rGO; white diamond (◇), ZnO; black triangle (▼), Ag. Data are representative of two separate experiments.

**Figure 3 ijms-20-05394-f003:**
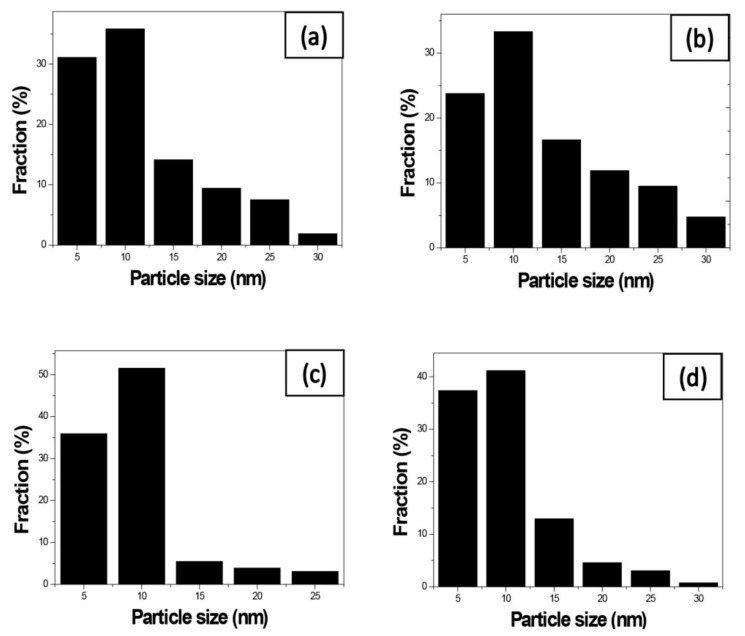
Size distribution of the nanocomposites. Size distribution of (**a**) 5.34% silver nanoparticle-embedded graphene oxide (Ag/rGO), (**b**) 7.49% Ag/rGO, (**c**) 6.85% zinc oxide nanoparticle-embedded graphene oxide (ZnO/rGO), and (**d**) 16.45% ZnO/rGO. Data are representative of two separate experiments.

**Figure 4 ijms-20-05394-f004:**
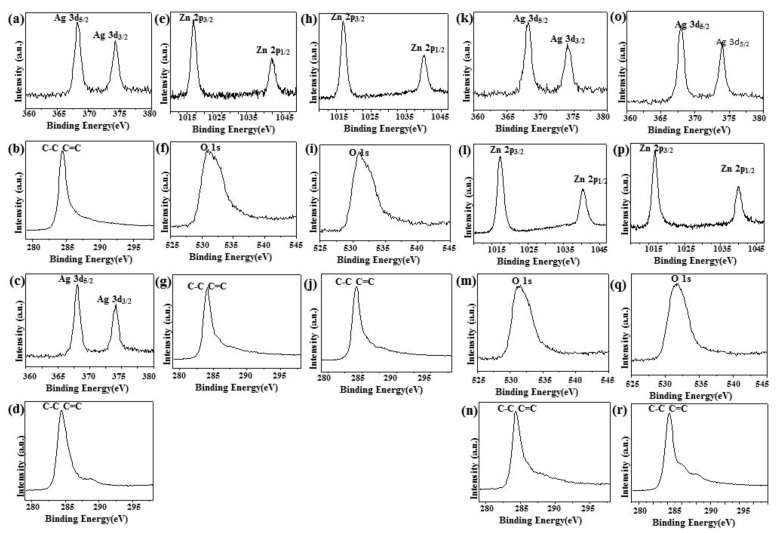
X-Ray photoelectron spectroscopy. X-Ray photoelectron spectroscopy (XPS) analysis of (**a**,**b**) 5.34% Ag/rGO, (**c**,**d**) 7.49% Ag/rGO, (**e**–**g**) 6.85% ZnO/rGO, (**h**–**j**) 16.45% ZnO/rGO, (**k**–**n**) 3.47/34.91% Ag/ZnO/rGO, and (**o**–**r**) 7.08/15.28% Ag/ZnO/rGO.

**Figure 5 ijms-20-05394-f005:**
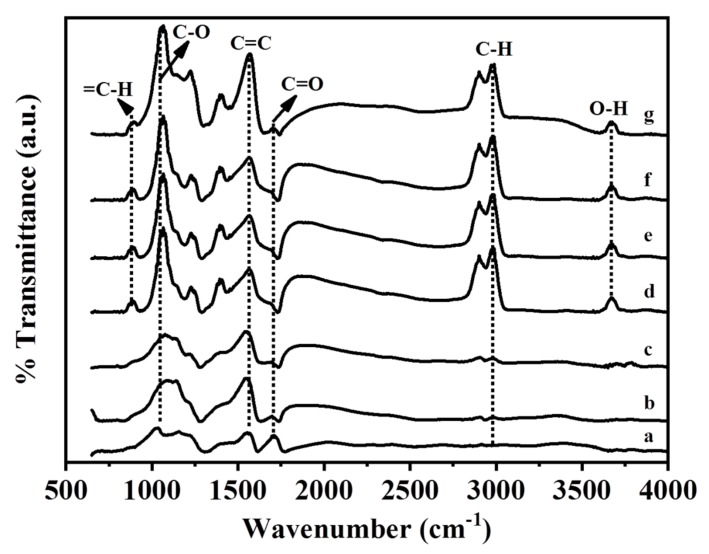
Fourier-transform infrared spectra analysis. (**a**) GO as the control, (**b**) 5.34% Ag/rGO, (**c**) 7.49% Ag/rGO, (**d**) 6.85% ZnO/rGO, (**e**) 16.45% ZnO/rGO, (**f**) 3.47/34.91%Ag/ZnO/rGO, and (**g**) 7.08/15.28% Ag/ZnO/rGO. Data are representative of two separate experiments.

**Figure 6 ijms-20-05394-f006:**
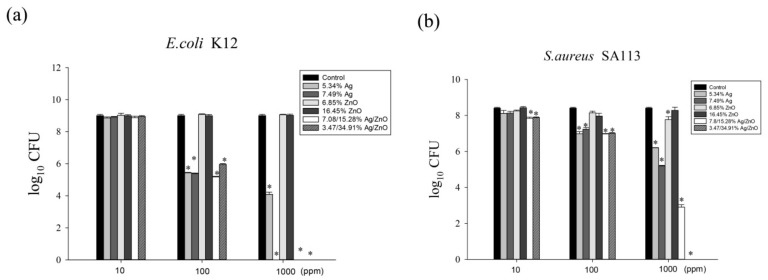
The antibacterial effects of Ag/rGO, ZnO/rGO and Ag/ZnO/rGO nanocomposites. (**a**) *E. coli* K12, (**b**) *S. aureus* SA113. Antibacterial effects of 5.34% Ag/rGO (medium light gray), 7.49% Ag/rGO (medium dark gray), 6.85% ZnO/rGO (light gray), 16.45% ZnO/rGO (dark gray), 3.47/34.91% Ag/ZnO/rGO (diagonal bar), and 7.08/15.28% Ag/ZnO/rGO (white bar) at concentrations of 10, 100, and 1000 ppm, against *Escherichia coli* K12 and *Staphylococcus aureus* SA113 strains grown in MHB medium at 37 °C. The control (black color) was without any treatment. Data are expressed as mean ± standard deviation of two separate experiments, with three replicates. Asterisk represents *p* < 0.05.

**Figure 7 ijms-20-05394-f007:**
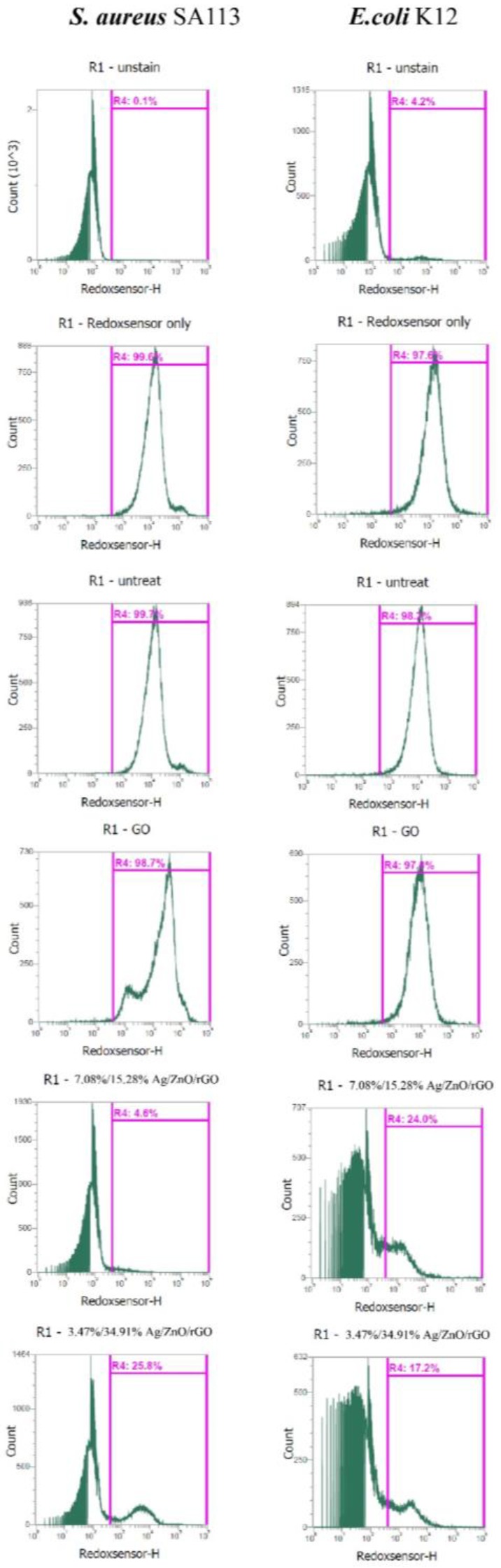
Analysis of RedoxSensor staining intensity in *E. coli* K12 and *S. aureus* SA113 strains. These two strains were grown in MHB supplemented with 100 ppm of 3.47/34.91% and 7.08/15.28% Ag/ZnO/rGO composites for 3 h, and the cells were then treated with RedoxSensor (green). The X-axis represents the RedoxSensor intensity in arbitrary units (a.u.) and the Y-axis represents the cell counts as measured by flow cytometry. PBS-only and unstained cells were used as the controls. RedoxSensor fluorescence is presented in a false green color. Data are representative of two separate experiments. PBS buffer only: addition of only PBS buffer (no cells or fluorescent dyes); unstained: untreated bacterial cells that did not stain with fluorescent dyes; PI only: untreated bacterial cells stained with PI dye only without Redox dye; Redox only: bacterial cells stained with Redox dye but not PI dye; Ag/ZnO/rGO: bacterial cells stained with both PI and Redox dyes at 100 ppm Ag/ZnO/rGO. All parameters were collected as logarithmic signals.

**Figure 8 ijms-20-05394-f008:**
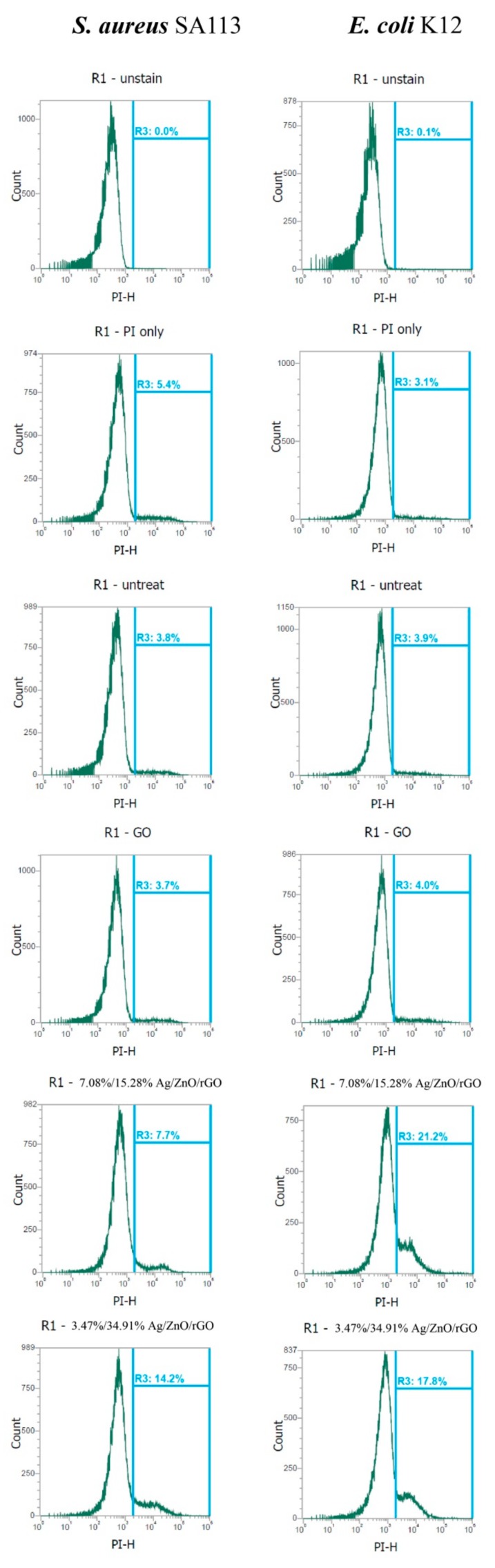
Analysis of propidium iodide (PI) staining intensity in *E. coli* K12 and *S. aureus* SA113 strains. These two strains were grown in MHB supplemented with 100 ppm of 3.47/34.91% and 7.08/15.28% Ag/ZnO/rGO composites for 3 h, and the cells were then treated with PI (green). The X-axis represents the PI intensity in arbitrary units (a.u.) and the Y-axis represents the cell counts as measured by flow cytometry. PBS-only and unstained cells were used as the controls. PI fluorescence is presented in a false blue color. PBS buffer only: addition of only PBS buffer (no cells or fluorescent dyes); unstained: untreated bacterial cells that did not stain with fluorescent dyes; PI only: untreated bacterial cells stained with PI dye only without Redox dye; Redox only: bacterial cells stained with Redox dye but not PI dye; Ag/ZnO/rGO: bacterial cells stained with both PI and Redox dyes at 100 ppm Ag/ZnO/rGO. All parameters were collected as logarithmic signals. Data are representative of two separate experiments.

**Table 1 ijms-20-05394-t001:** Quantitative XPS measurement of 5.34% Ag/rGO, 7.49% Ag/rGO, 6.85%ZnO/rGO, 16.45% ZnO/rGO, 3.47/34.91% Ag/ZnO/rGO, and 7.08/15.28% Ag/ZnO/rGO.

Synthesized Sample wt% (NPs/rGO)	Percentages of Element (wt%)	XPS Measured: wt% (NPs/rGO)
7.5% Ag/rGO	C: 72.61, Ag: 5.34, others: 22.05	5.34% Ag
12% Ag/rGO	C: 72.54, Ag: 7.49, others: 19.97	7.49% Ag
7.5% ZnO/rGO	C: 68.71, Zn: 5.51, O: 18.32, others: 7.46	6.85% ZnO
12% ZnO/rGO	C: 61.05, Zn:13.22, O: 17.78, others: 7.95	16.45% ZnO
7.5%/7.5% Ag/ZnO/rGO	C: 58.06, Ag: 7.08, Zn: 12.28, O: 19.73, others: 2.85	7.08%/15.28% Ag/ZnO
3%/12% Ag/ZnO/rGO	C: 39.12, Ag: 3.47, Zn: 28.05, O: 19.26, others: 10.1	3.47%/34.91% Ag/ZnO

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
