# Peer review of "Antibacterial Property of Composites of Reduced Graphene Oxide with Nano-Silver and Zinc Oxide Nanoparticles Synthesized Using a Microwave-Assisted Approach"

_ijms, 2019, doi:10.3390/ijms20215394_

Round 1

Reviewer 1 Report

In the present manuscript, microwave-assisted chemical reduction was used to deposit nano-silver and zinc oxide NPs (Ag and ZnO NPs) on the surface of reduced GO (rGO). Manuscript worth consideration; however, there are suggestions to improve the manuscript before it can get accepted for publications as are outlined below:

Novelty of this work needs to be demonstrated as Ag/ZnO/rGO nanocomposites in bacterial detection and sterilization are well established. Results section line 72, Authors found the integrated area percentages of Ag/rGO, ZnO/rGO, and Ag/ZnO/rGO to be 5.34%, 7.49%, 6.85%,16.45%, 3.47/34.91%, and 73 7.08/15.28%, respectively. Are these rations happened randomly or intentionally? This needs to be explained properly, and also what effect changing these rations may have. Figure1, scale of SEM is not readable, also need to explain the differences between each SEM graph How many replicates were involved? Statistical analysis like ANOVA is essential. Section 2.4. The Ag/ZnO/rGO nanocomposite inhibited bacterial growth, what was the control sample? Explain why 10, 100 and 1000 ppm for antimicrobial tests have been chosen? Please in M&M make sure to cite the methods that you have adopted them in your work. In M&M, explain from where E. coli K12 and S. aureus SA113 are purchased I hardly see any 2019/2018 publications, make sure you cite and discuss the recent papers in your discussion.

Reviewer 2 Report

This is an interesting topic of great biomedical interest. However this biomedical transcendence is not reflected in the manuscript. In addition, some adjustments should be maid:

The title is too long Figure captions for figures 7 and 8 are too long Discussion of results is scarce. Indeed, they widely relate the experiments of other authors, but they do slightly relate those results with the original ones presented in the manuscript. A wider and more detailed description of the possible biomedical applications is missing

Reviewer 3 Report

The provided SEM images are not clear to claim morphology. Also, the scale bars in the SEM images Fig. 1 are not clear. Need to provide the better quality images. X-axis values are not in the visible range for Fig.2. How authors extracted particle sizes, need to explain? Need to cite the following recent antibacterials based articles including 10.1016/j.ijbiomac.2018.11.251; 10.3390/nano7110363; 10.1016/j.ijbiomac.2019.07.120

Round 2

Reviewer 1 Report

Authors made the suggested changes and I support acceptance 

Author Response

thanks reveiwer's comments, we appreciated it.

Reviewer 3 Report

Why authors indexed C-C C=C in the carbon XPS peak. It is better to review other literature and provide the exact atom bonding instead of provided one. Also, some low intensity peaks are observed for carbon and oxygen binding energies. Should be provided the fitted curves and indexed their positions for all the carbon and oxygen XPS. How was the statistical ANOVA analysis was performed i.e., type of software and input details. Should be included their details in the experimental part. What has concluded from the statistical ANOVA, need to elaborate in the manuscript? What is arbitrary unit in the antibacterial activity, need to provide the correct unit or explanation? Need to provide all the JCPDS standard details in the discussion. Authors provided in the response for Reviewer query related to the particles size “The distribution in particle size of each sample was obtained by counting more than 100 nanoparticles on the SEM setup”. But in the manuscript, authors included as “The morphology, microstructure, and particle size of the Ag/ZnO/GO NPs were investigated by high-resolution transmission electron microscopy (HR-TEM; H-7500; Hitachi, Tokyo, Japan).” Which is true. In addition, TEM results are not provided in the manuscript. Authors claimed in the revision, “size distribution of each sample was obtained by counting more than 100 nanoparticles in the field-emission scanning electron microscopy micrograph.” But, there is no clear grain sizes. Need to provide the higher magnification images. The revised figure 1 quality is not good. Need to provide high quality image to satisfy their claim and discussion. Also authors provided “The morphology, microstructure, and particle size of the Ag/ZnO/GO NPs were investigated by high-resolution transmission electron microscopy (HR-TEM; H-7500; Hitachi, Tokyo, Japan)” under the subhead 4.5. XRD….. Authors should recheck. The sections 4.4, 4.5 and 4.6 are entirely confusing and collapsing. Need to rewrite carefully. What has author intended to claim by the below lines in 331-335, page 13. “The C 1s (284.5 eV) peak was used as the calibration standard for the wide-region spectra of these samples with different C–C valences. XPS signals of the above species were recorded with a cylindrical mirror analyzer. The fractions of these samples with different C–C valences were calculated by their integrated peak areas.” The figure caption 7 and 8 are need to recheck. Example authors mentioned “The X-axis represents the ----- intensity in arbitrary units”, is it right?

Round 3

Reviewer 3 Report

Authors have responded for all the queries satisfactorily. However, they need to change the characterization details as a Minor revision for the publication.

Authors have responded for all the queries satisfactorily. However, they need to change the characterization details as a Minor revision for the publication. The comments are follows.

Authors used to describe the FESEM details under the section of 4.5. XRD characterization. Similarly authors used to define XRD based analysis under the section of 4.6. FESEM.

4.5. XRD characterization

The XRD patterns of the synthesized GO and Ag NP- and ZnO NP-doped GO composites were recorded at a scan rate of 4 (2θ)/min using monochromatic Cu Kα radiation (MXP18; MAC Science Co., Tokyo, Japan) at 30 kV and 20 mA. The recorded specific peak intensity and 2θ values were further identified using a database system (JCPDS). The morphology, microstructure, and particle size of the Ag/ZnO/GO NPs were investigated by a field emission scanning electron microscope (Rigaku, Tokyo, Japan).

4.6. Field emission scanning electron microscopy

We performed electron microscopy using a RU-H3R field emission scanning electron microscope (Rigaku, Tokyo, Japan) to characterize the prepared samples in terms of their morphology, structure, and elemental composition. A Xay-ray diffraction (Shimadzu labx XRD-6000) spectroscope equipped with Cu-K α radiation emitter (1.5405 A0), was used to characterize crystal structures of carbon material. The crystallite and solid-phase structures of GO and Ag NP- and ZnO NP-doped GO composites were characterized using monochromatic Cu Kα radiation at 1.5405 Å, 30 kV, and 20 mA. The maximal output of X-ray generator is 3 kV and the dimension of XRD equipment is W900×D700×H1600.

So, authors should interchange their details to the appropriate subheading or should be use the common section. Also, authors should be avoid the repeatable instrument details.

Author Response

We thank reviewer’s comments and the revised 4.5, and 4. 6 sections have been changed in text.

Authors have responded for all the queries satisfactorily. However, they need to change the characterization details as a Minor revision for the publication. The comments are follows.

Authors used to describe the FESEM details under the section of 4.5. XRD characterization. Similarly authors used to define XRD based analysis under the section of 4.6. FESEM.

4.5. XRD characterization

The XRD patterns of the synthesized GO and Ag NP- and ZnO NP-doped GO composites were recorded at a scan rate of 4 (2θ)/min using monochromatic Cu Kα radiation (MXP18; MAC Science Co., Tokyo, Japan) at 30 kV and 20 mA. The recorded specific peak intensity and 2θ values were further identified using a database system (JCPDS). The morphology, microstructure, and particle size of the Ag/ZnO/GO NPs were investigated by a field emission scanning electron microscope (Rigaku, Tokyo, Japan).

4.6. Field emission scanning electron microscopy

We performed electron microscopy using a RU-H3R field emission scanning electron microscope (Rigaku, Tokyo, Japan) to characterize the prepared samples in terms of their morphology, structure, and elemental composition. A Xay-ray diffraction (Shimadzu labx XRD-6000) spectroscope equipped with Cu-K α radiation emitter (1.5405 A0), was used to characterize crystal structures of carbon material. The crystallite and solid-phase structures of GO and Ag NP- and ZnO NP-doped GO composites were characterized using monochromatic Cu Kα radiation at 1.5405 Å, 30 kV, and 20 mA. The maximal output of X-ray generator is 3 kV and the dimension of XRD equipment is W900×D700×H1600.

Ans: Revised as following:

4.5. XRD characterization

A X-ray diffraction (XRD, Shimadzu labx XRD-6000) spectroscope was used to characterize crystalline structures of carbon composites. The XRD patterns of the synthesized GO and Ag NP- and ZnO NP-doped GO composites were recorded at a scan rate of 4 degree/min using monochromatic Cu-Kα radiation (MXP18; MAC Science Co., Tokyo, Japan) at 30 kV and 20 mA. The recorded specific peak intensity and 2θ values were further identified using a database system (JCPDS).

4.6. Field emission scanning electron microscopy

The morphology, dispersion, and particle size of the Ag/ZnO NPs on rGO support were investigated by a field emission scanning electron microscope (FE-SEM, Rigaku, Tokyo, Japan). We also performed electron microscopy using a RU-H3R FE-SEM to characterize the prepared samples in terms of their structure and elemental composition.